# Structural basis for *Sarbecovirus* ORF6 mediated blockage of nucleocytoplasmic transport

Xiaopan Gao [1,5], Huabin Tian[2,5], Kaixiang Zhu[1,5], Qing Li[2,3], Wei Hao[1], Linyue Wang[1], Bo Qin[1], Hongyu Deng [2,3] ✉ & Sheng Cui [1,4] ✉

The emergence of heavily mutated SARS-CoV-2 variants of concern (VOCs) place the international community on high alert. In addition to numerous mutations that map in the spike protein of VOCs, expression of the viral accessory proteins ORF6 and ORF9b also elevate; both are potent interferon antagonists. Here, we present the crystal structures of Rae1-Nup98 in complex with the C-terminal tails (CTT) of SARS-CoV-2 and SARS-CoV ORF6 to 2.85 Å and 2.39 Å resolution, respectively. An invariant methionine (M) 58 residue of ORF6 CTT extends its side chain into a hydrophobic cavity in the Rae1 mRNA binding groove, resembling a bolt-fitting-hole; acidic residues flanking M58 form salt-bridges with Rae1. Our mutagenesis studies identify key residues of ORF6 important for its interaction with Rae1-Nup98 in vitro and in cells, of which M58 is irreplaceable. Furthermore, we show that ORF6-mediated blockade of mRNA and STAT1 nucleocytoplasmic transport correlate with the binding affinity between ORF6 and Rae1-Nup98. Finally, binding of ORF6 to Rae1-Nup98 is linked to ORF6-induced interferon antagonism. Taken together, this study reveals the molecular basis for the antagonistic function of *Sarbecovirus* ORF6, and implies a strategy of using ORF6 CTT-derived peptides for immunosuppressive drug development.

Emerging genetic variants of severe acute respiratory syndrome coronavirus 2 (SARS-CoV-2) present a formidable challenge for containing the COVID-19 pandemic. Variant of concern (VOC) refers to SARS-CoV-2 isolates that exhibit enhanced transmissibility[1], virulence[2] and immune evasion[3,4], cause severer disease or reduce the effectiveness of current diagnostics, vaccines,and therapeutics. The Omicron variant (B1.1.529) was recently designated as a top priority VOC because of an unprecedented large number of mutations that cluster in the spike protein[5]. Previous studies demonstrate that VOCs not only harbor mutations in key residues of the spike protein for evading neutralizing

antibodies, but also elevate the suppression of innate immunity during infection by upregulating subgenomic RNA and protein levels of innate immune antagonists, such as SARS-CoV-2 ORF9b and ORF6[6].

ORF9b and ORF6 are accessory proteins of *Sarbecovirus* including SARS-CoV-2 and SARS-CoV[7–9]. Whereas ORF9b suppresses interferon (IFN) production by targeting the mitochondrial multifunctional adapter TOM70[10–12], ORF6 dampens antiviral immune responses by blocking bidirectional nucleocytoplasmic transport of cellular mRNAs and proteins[13]. SARS-CoV-2 ORF6 antagonizes antiviral immunity more efficiently than SARS-CoV ORF6, providing a possible explanation for

[1]NHC Key Laboratory of Systems Biology of Pathogens, Institute of Pathogen Biology, Chinese Academy of Medical Sciences and Peking Union Medical College, Beijing 100730, P. R. China. [2]CAS Key Laboratory of Infection and Immunity, CAS Center for Excellence in Biomacromolecules, Institute of Biophysics, Chinese Academy of Sciences, Beijing 100101, China. [3]University of Chinese Academy of Sciences, Beijing 100049, China. [4]Sanming Project of Medicine in Shenzhen, National Clinical Research Center for Infectious Diseases, Shenzhen Third People's Hospital, Southern University of Science and Technology, Shenzhen, China. [5]These authors contributed equally: Xiaopan Gao, Huabin Tian and Kaixiang Zhu. ✉e-mail: hydeng@moon.ibp.ac.cn; cui.sheng@ipb.pumc.edu.cn

asymptomatic infection or delayed symptom onset in SARS-CoV-2-infected patients[14].

*Sarbecovirus* ORF6 targets the Rae1-Nup98 complex, a component on the cytoplasmic face of the nuclear pore complex (NPC)[13,15,16]. Previous studies suggest that the C-terminal tail (CTT) of ORF6 is crucial for its interaction with the Rae1-Nup98 complex and immune antagonistic activity[14,17]. A methionine flanked by acidic residues in the ORF6 CTT is fundamental to its functions[16]. Nonetheless, the molecular mechanism underlying ORF6-mediated blockade of nucleocytoplasmic trafficking and suppression of antiviral immunity remains elusive.

ORF6 is reminiscent of the vesicular stomatitis virus (VSV) matrix protein (M) and herpesviruses (Kaposi's sarcoma-associated herpesvirus or murine gammaherpesvirus 68 [MHV-68]) ORF10, both of which target cellular mRNA nuclear export[18–20]. Of note, both VSV M and MHV-68 ORF10 contain a special methionine that is essential for binding the Rae1-Nup98 complex and for retaining mRNA in the nucleus[15,19,21]. This methionine is located at the N-terminal extension (NTE) of VSV M (M51), while that of MHV-68 ORF10 (M413) is at the CTT. Crystallographic investigation reveals that both VSV M M51 and MHV-68 ORF10 M413 occupy the same hydrophobic cavity in the Rae1-Nup98 complex, suggesting that the cavity is an "Achilles heel" of the NPC targeted by a variety of viruses[19,22].

In this study, we determine the X-ray crystal structures of Rae1-Nup98 in complex with the C-terminal tails (CTT) of SARS-CoV-2 and SARS-CoV ORF6 to 2.85 Å and 2.39 Å resolution, respectively. The structure reveals that M58 residue of ORF6 CTT extends its side chain

into a hydrophobic cavity in the Rae1 mRNA binding groove, resembling a bolt-fitting-hole, acidic residues flanking M58 form salt-bridges with Rae1. We show that key residues of ORF6 are important for its interaction with Rae1-Nup98 in vitro and in vivo. Furthermore, we demonstrate that ORF6-mediated blockade of mRNA and STAT1 nucleocytoplasmic transport correlate with the binding affinity between ORF6 and Rae1-Nup98. In summary, our study reveals the molecular basis for the antagonistic function of *Sarbecovirus* ORF6 and implies a strategy of using ORF6 CTT-derived peptides for immunosuppressive drug development.

## Results and discussion

To gain structural insight into *Sarbecovirus* ORF6-mediated nuclear transport inhibition and innate immunity suppression, we sought to determine the crystal structure of the ORF6-Rae1-Nup98 complex. While we co-expressed a Rae1 fragment (residues 31–368) bound by the Gle2/Rae1-binding sequence (GLEBS, residues 157–213) of Nup98 in insect cells (Fig. 1a–c), ectopic expression of ORF6 proteins in insect cells or bacteria was unsuccessful, probably due to their cytotoxicity and/or the presence of a membrane anchoring helix at the NTE[23], which might undermine protein stability. To circumvent this problem, we synthesized a panel of peptides covering different regions in the ORF6 CTT (Fig. 1d), and measured the binding affinity of these ORF6-derived peptides to Rae1-Nup98 using isothermal titration calorimetry (ITC). All peptides (peptide C1-C4, Fig. 1e–h and Supplementary Table 1) containing M58 and the surrounding acidic residues bound Rae1-Nup98 with nanomolar affinity ($K_d = 240-440$ nM). Next, we

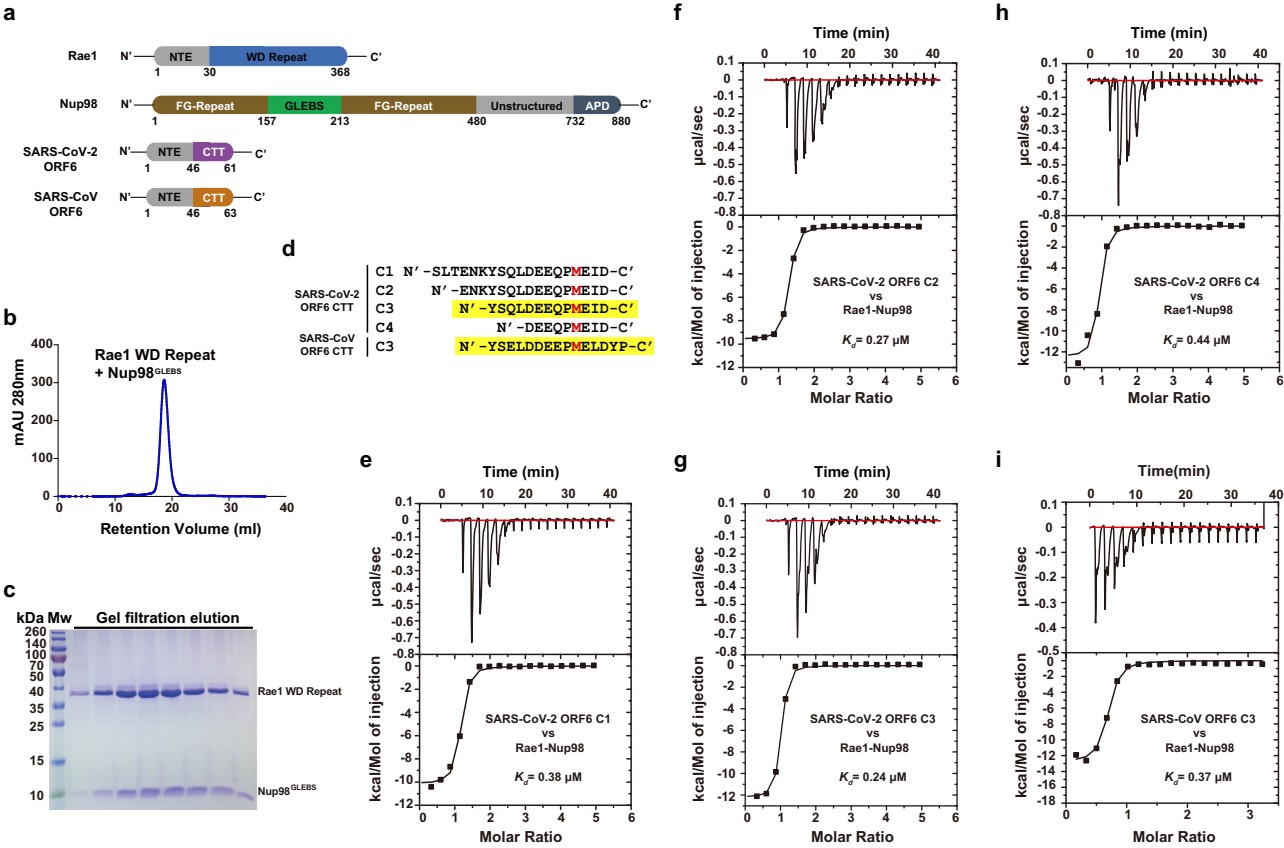

**Fig. 1 | Peptides containing the C-terminal tails of SARS-CoV-2 and SARS-CoV ORF6 proteins interact with the Rae1-Nup98 complex at nanomolar affinity. a** Diagrams of domain organization of Rae1, Nup98 and *Sarbecovirus* ORF6 proteins with annotations. **b** Final purification step of Rae1 WD Repeat−Nup98$^{GLEBS}$ complex. Size-exclusion chromatography profile of Rae1-Nup98 complex. **c** SDS-PAGE analysis of the eluate from the size-exclusion chromatography shown in **b**. Results

shown are representative of three independent experiments. **d** Sequence of SARS-CoV-2 and SARS-CoV ORF6 CTT peptides (C1-C4); key methionine M58 is highlighted in red; two C3 peptides that co-crystallized with Rae1-Nup98 complex are highlighted with yellow background. **e−h** ITC titrations between SARS-CoV-2 ORF6 C1-C4 peptides and Rae1-Nup98. **i** ITC experiment titration between SARS-CoV ORF6 C3 peptide and Rae1-Nup98. Source data are provided as a Source data file.

mixed Rae1-Nup98 with those peptides (molar ratio = 1:4) for crystallization, and obtained crystals of Rae1-Nup98 in the presence of peptide C3 (N'- $_{49}$YSQL*DEE*QPM*EID*$_{61}$-C', SARS-CoV-2 ORF6 C3). Using a similar protocol, we investigated the interaction of a peptide derived from an equivalent region of the SARS-CoV ORF6 CTT (N'-$_{49}$YSEL*DEE*PM*ELD*YP$_{63}$-C', SARS-CoV ORF6 C3) with Rae1-Nup98. The binding affinity of SARS-CoV-2 ORF6 C3 to Rae1-Nup98 ($K_d$ = 240 nM) was slightly higher than that of SARS-CoV ORF6 C3 ($K_d$ = 370 nM, Fig. 1g, i). Next, we co-crystallized Rae1-Nup98 with each of these two peptides. The crystals of SARS-CoV-2 ORF6 C3-Rae1-Nup98 and SARS-CoV ORF6 C3-Rae1-Nup98 diffracted the X-ray to 2.85 Å and 2.39 Å, respectively. We determined both crystal structures by molecular replacement (searching model PDB id: 4OWR). Statistics and parameters of data collection and structure refinement are summarized in Supplementary Table 2.

Our crystal structures revealed that both SARS-CoV-2 ORF6 C3 and SARS-CoV ORF6 C3 peptides target a positively charged groove (also known as a putative RNA binding groove, Fig. 2a, b) on the rim of Rae1 β-propellers, and we did not find contact between the peptides and Nup98$^{GLEBS}$. We calculated the composite omit maps (with anneal method) for both structures. The maps clearly delineate nine residues in SARS-CoV-2 ORF6 C3 and ten residues in SARS-CoV ORF6 C3 (Fig. 2a, b). While the last residue of SARS-CoV-2 ORF6 D61 was clearly visible in the electron density map, two additional C-terminal residues Y62 and P63 following the D61 of SARS-CoV ORF6 were barely visible. Furthermore, the C-terminus of SARS-CoV ORF6 points away from the RNA binding groove of Rae1, which suggests that the two extra C-terminal residues of SARS-CoV ORF6 are not essential for binding.

SARS-CoV-2 and SARS-CoV ORF6 C3 peptides bind Rae1 with the same orientation (Fig. 2c, d). In both structures, residue M58 at the ORF6 CTT extends its hydrophobic side chain into a deep hydrophobic cavity in the RNA-binding groove of Rae1, resembling a bolt-fitting-hole. The acidic residues (glutamate and aspartate) flanking M58 form salt-bridges with positively charged residues (lysine and arginine) in the RNA-binding groove of Rae1. In addition, we observed several mainchain-mediated hydrogen bonds between the ORF6 CTT and Rae1, which strengthen their interactions. A proline immediately upstream of M58 introduces a bend to the peptides, stabilizing it in an ideal conformation for hydrogen bond interactions with Rae1. Specifically, the cis-configuration of P57 allows for two hydrogen-bonds between the ORF6 residue 56 (SARS-CoV-2 Q56 or SARS-CoV E56) and Rae1 K307, and between the ORF6 M58 and Rae1 R305 (Fig. 2c, d). P57 is not only invariant in *Sarbecovirus* ORF6 (Supplementary Fig. 1), but also present in MHV-68 ORF10 (Fig. 2e). M413 of MHV-68 ORF10 CTT is crucial for the interaction with the RNA-binding groove of Rae1-Nup98, and this methionine is preceded by P412. By contrast, VSV M binds to the opposite side of Rae1-Nup98 with its NTE, and a proline adjacent to M51 is unavailable. Collectively, our analyses suggest that M58 of the ORF6 CTT is essential for binding Rae1.

Given that different viral proteins target the same cavity in Rae1 (constituted by F255, F257, W300 and R305) with a methionine residue, we denoted this cavity as the M-cavity. Among the 20 essential proteogenic amino acids, methionine is the only one with non-branched and non-aromatic hydrophobic side chain, suggesting that the M-cavity strictly selects for size and flexibility of the hydrophobic side chains, and only methionine matches the selection. Superimposing the two structures determined in this study onto the structures of VSV M-Rae1-Nup98 and MHV-68 ORF10-Rae1-Nup98 demonstrates that the interaction between the crucial methionine and the M-cavity are similar (Fig. 2e, right). During our manuscript in revision, another group published the crystal structures of ORF6 CTT-Rae1-Nup98 complex from SARS-CoV-2 and SARS-CoV[24]. Comparing our crystal structures with theirs revealed similar features, including the specific interaction between ORF6 M58 and the M-cavity in Rae1-Nup98. Structural superimposition gave a root mean square deviation

(RMSD) of 0.37 Å over 361 aligned Cα atoms for two SARS-CoV-2 ORF6 CTT-Rae1-Nup98 structures and 0.22 Å over 340 aligned Cα atoms for two SARS-CoV ORF6 CTT-Rae1-Nup98 structures (Supplementary Fig. 2). The specificity of the M-pocket for methionine is also supported by other structural investigations. The structure of the Rae1-Nup98 complex alone (PDB id: 3MMY) shows that the M-pocket of Rae1 is occupied by an irrelevant M17 of an adjacent Rae1 molecule in crystal lattice[25]. A cell-cycle arrest protein Bub3 in yeast is a structural homolog of Rae1, which also comprises a 7-bladed β-propeller (PDB id: 4BL0). Bub3 harbors a deep cavity constituted by F236, F238, W278 and R283, identical to the M-pocket in Rae1. The M-cavity of Bub3 is occupied by M169 at the MELT repeats[26], which is similar to viral protein-Rae1 interactions.

To identify the molecular determinants governing the interaction between *Sarbecovirus* ORF6 and Rae1-Nup98, we synthesized a selection of peptides derived from different viral proteins and compared their binding affinity to Rae1-Nup98 using ITC (Supplementary Table 1, 3). One major difference between SARS-CoV-2 ORF6 and SARS-CoV ORF6 is the lack of a C-terminal Y62-P63 (YP) in the former. Our crystallographic studies suggested that the C-terminus YP of SARS-CoV ORF6 did not directly contact the Rae1-Nup98 complex (Fig. 2b, d), which is consistent with ITC results showing that SARS-CoV-2 ORF6 C3 and SARS-CoV ORF6 C3 peptides bound Rae1-Nup98 with similar affinity (Fig. 1g, i). To investigate the contribution of YP in binding, we measured the binding affinity of a SARS-CoV-2 ORF6 C3 harboring a YP extension (SARS-CoV-2 ORF6 C3 + YP) to Rae1-Nup98. Indeed, adding a YP extension did not affect binding affinity dramatically ($K_d$ = 0.20 µM, Fig. 3a and Supplementary Fig. 3a). Conversely, removing YP from the C-terminus of SARS-CoV ORF6 C3 (SARS-CoV ORF6 C3 -YP) increased the binding affinity by 3.4-fold ($K_d$ = 0.11 µM, Fig. 3a and Supplementary Fig. 3b), indicating that YP might negatively modulated the binding in SARS-CoV ORF6. The different role of YP in the binding of SARS-CoV-2 ORF6 and SARS-CoV ORF6 C3 peptides to the Rae1-Nup98 complex implies that YP might affect binding in synergy with other residues specific to SARS-CoV ORF6, but not to SARS-CoV-2 ORF6.

We further carried out systematic mutagenesis to determine the role of individual residue of SARS-CoV-2 ORF6 CTT in Rae1-Nup98 binding. Mutations of the acidic residues D53A, E54A, E55A, E59A, D61A and P57A adjacent to M58 moderately reduced the binding affinity to Rae1-Nup98 by 2.4–11.5 folds (Fig. 3a, Supplementary Fig. 3c–e, Fig. 3b–d). By contrast, altering residues I60 and M58 of SARS-CoV-2 ORF6 C3 led to a greater loss of binding affinity. Whereas the I60A mutation reduced binding affinity by -17.5 folds (Fig. 3a, e), the M58A mutation abolished the binding affinity (Fig. 3a, f), confirming the essential role of M58 in binding. To further investigate the selectivity of M58 in binding Rae1-Nup98, we replaced M58 with an arginine or a leucine. M58R mutation abolished the binding, which confirms that the hydrophobic side chain is selected by the M-cavity (Supplementary Fig. 3f). M58L still caused -43 folds decline in binding affinity ($K_d$ = 10.37 µM), even though leucine has similar hydrophobicity as methionine[27] (Supplementary Fig. 3g). Possibly, the branched side chain of leucine could introduce steric hindrance for accessing the M-pocket. Furthermore, while SARS-CoV-2 ORF6 C3 lacking the last four residues $_{58}$-MEID-$_{61}$ did not bind Rae1-Nup98, a peptide lacking the last three residues $_{59}$-EID-$_{61}$ but retaining M58 showed a weak binding affinity ($K_d$ = 87.71 µM, Supplementary Fig. 3h, i), indicating the requirement of Rae1 M-cavity for methionine. Collectively, these results provide direct evidence that the Rae1 M-cavity strictly selects methionine for binding.

We further compared the binding affinity of peptides derived from MHV-68 ORF10 CTT and VSV M NTE with that of ORF6-derived peptides using ITC. MHV-68 ORF10 CTT and VSV M NTE bound Rae1-Nup98 with lower affinity ($K_d$ = 5.71 µM and 116.95 µM, Fig. 3g, h) than ORF6-derived peptides. A plausible explanation is that VSV M and MHV-68 ORF10 are larger than ORF6, and they both contain other

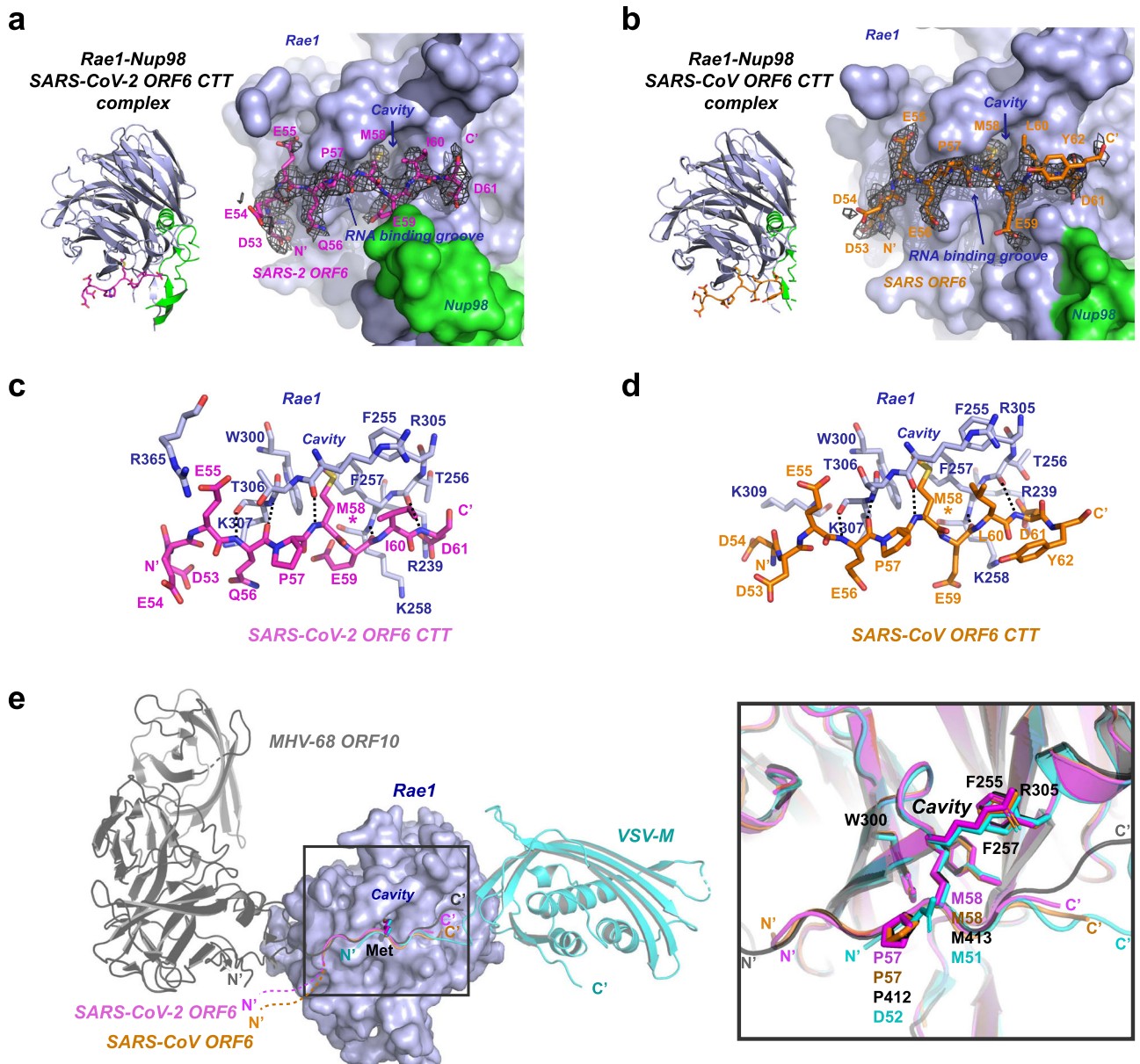

**Fig. 2 | Crystal structures of SARS-CoV-2 and SARS-CoV ORF6 CTT peptides complexed with Rae1-Nup98. a** Left, ribbon model of the SARS-2 ORF6 C3 peptide complexed with Rae1-Nup98. Right, magnified view of the peptide-protein interaction. The Rae1 (blue)−Nup98 (green) complex is shown with molecular surface, the bound SARS-CoV-2 ORF6 C3 is shown with magenta stick model; a composite omit map associated with the peptide is shown with black meshes. **b** Left, ribbon model of the SARS-CoV ORF6 C3 peptide complexed with Rae1-Nup98. Right, magnified view of the peptide-protein interaction. The Rae1 (blue)−Nup98 (green) complex is shown with molecular surface, and the bound SARS-CoV ORF6 C3 is shown with orange stick model; a composite omit map associated with the peptide is shown with black meshes. **c** Detailed interactions between Rae1 (blue stick model) and the SARS-CoV-2 ORF6 C3 peptide (magenta stick model). Residues involved in the interaction are labeled, and intermolecular hydrogen bonds are shown with the dashed lines. **d** Detailed interactions between Rae1 (blue stick model) and the SARS-CoV ORF6 C3 peptide (orange stick model). Residues involved in the interaction are labeled, and intermolecular hydrogen bonds are shown with the dashed lines. **e** Left, superimposition of *Sarbecovirus* ORF6-CTT-Rae1-Nup98 complexes with VSV M-Rae1-Nup98 and MHV68 ORF10-Rae1-Nup98 complexes. While Rae1 is shown with blue molecular surface, all viral proteins are shown with ribbon model. SARS-CoV-2 ORF6 C3 peptide is colored magenta, SARS-CoV ORF6 C3 peptide orange, MHV-68 ORF10 gray, and VSV M cyan. Right, magnified view of the boxed area on the left panel. The key methionines and residues forming the deep pocket on Rae1 are shown with the stick model with labels.

domains that interact with Rae1-Nup98 in addition to their CTT or NTE. Consistent with our ITC results, recent papers demonstrated that the ORF6-mediated innate immunity suppression is largely dependent on the ORF6 CTTs[14,17].

Recent studies showed that SARS-CoV-2 harboring Q56E in ORF6 exhibited elevation in anti-IFN activity[14]. To reveal the mechanism behind this phenomenon, we measured the binding affinity of SARS-CoV-2 ORF6 C3 containing Q56E to Rae1-Nup98 (Fig. 3a,

Supplementary Fig. 3j). Whereas the Q56E mutation increased binding affinity to Rae1-Nup98 by ~6-folds, the Q56A mutation reduced the binding affinity with Rae1-Nup98 by ~2-fold (Fig. 3a, i). From a structural perspective, the Q56 of SARS-CoV-2 ORF6 lies within the salt-bridging range with the K307 of Rae1; therefore, Q56E could enhance electrostatic interaction between SARS-CoV-2 ORF6 and Rae1 (Fig. 2c). Thus, our results support that the binding affinity of ORF6 CTT to Rae1-Nup98 correlate with its anti-IFN activity. Of note, SARS-CoV ORF6

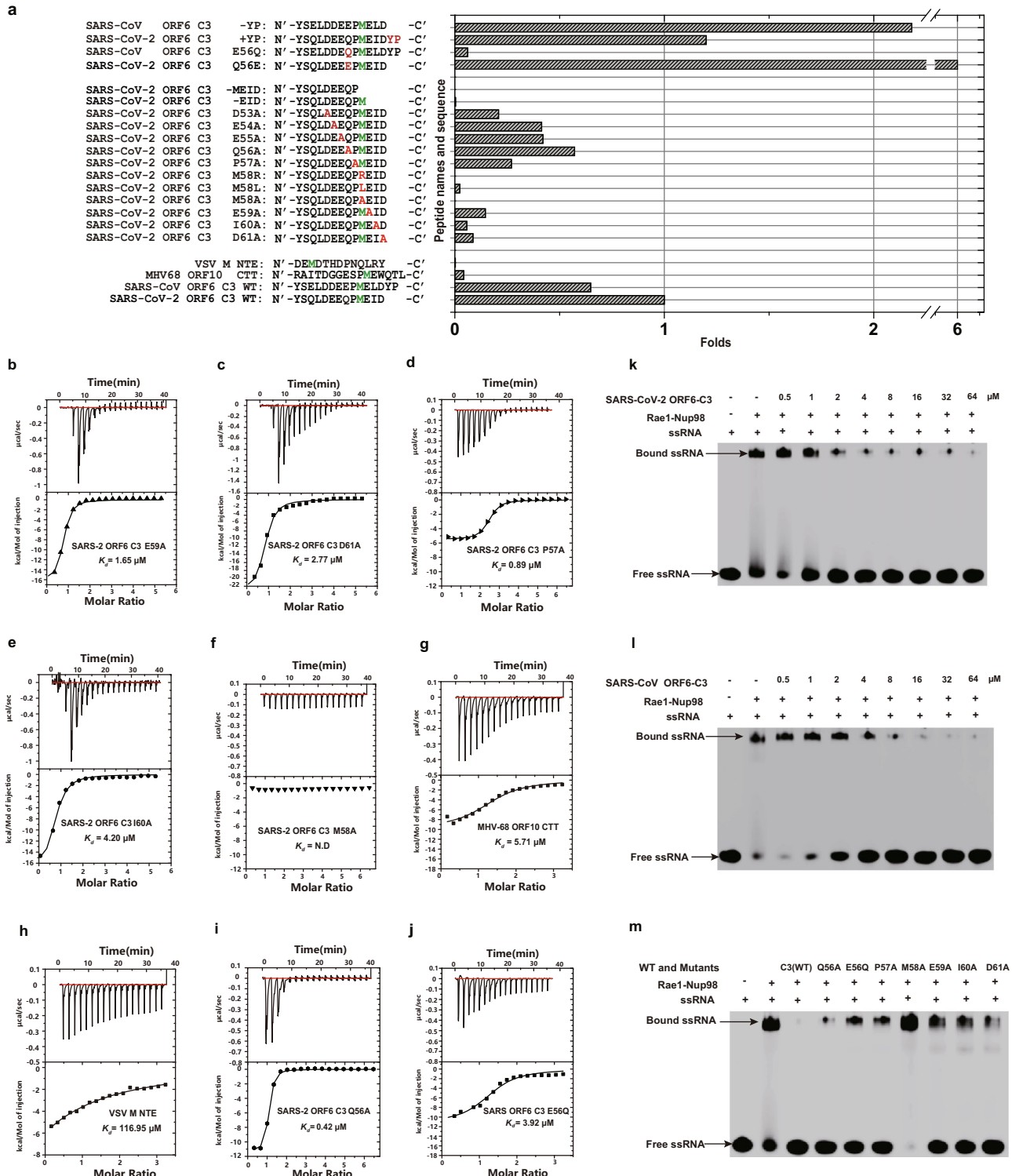

**Fig. 3 | Investigation of the ORF6-derived peptides binding to the Rae1-Nup98 complex and their inhibition of RNA binding with Rae1-Nup98. a** Histogram showing binding affinity of a selection of peptides derived from the CTT or NTE of different proteins to the Rae1-Nup98 complex. Key residues are highlighted in red and green. While the y axis indicates names and sequence of different peptides, the *x*-axis indicates the folds (folds of binding affinity) as compared with the affinity of SARS-CoV-2 ORF6 C3 peptide ($K_d = 0.24\,\mu M$). Thus, the folds of binding affinity were calculated: $0.24\,\mu M / K_d$ of the indicated peptide. **b–j** ITC experiments showing the interaction between each peptide and the Rae1-Nup98 complex. The binding affinity measured from each experiment is indicated. **k, l** EMSAs showing the ability of *Sarbecovirus* ORF6 CTTs (C3 peptide) to disrupt ssRNA binding with

Rae1-Nup98. After pre-incubating the FAM labeled 10-mer poly(U) ssRNA with Rae1-Nup98 complex, increasing amounts (0–64 μM) of SARS-CoV-2 or SARS-CoV ORF6 C3 peptides were added to the mixtures to allow competition with the ssRNA. The resulting mixtures were analyzed by native-PAGE and visualized using Typhoon scanner. This experiment was performed in three independent experiments, with similar results. **m** Mutations in SARS-CoV-2 or SARS-CoV ORF6 CTTs (indicated on top of the gel) peptide impaired its ability to compete with ssRNA binding for Rae1-Nup98. Concentrations of all peptides were 64 μM. Mutations were, from left to right, SARS-CoV-2 Q56A, SARS-CoV E56Q, SARS-CoV-2 P57A, M58A, E59A, I60A and D61A. This experiment was performed in three independent experiments, with similar results. Source data are provided as a Source data file.

contains E56, presenting another key difference from SARS-CoV-2 ORF6. We found that SARS-CoV ORF6 C3 harboring E56Q lost ~10.6-folds binding affinity to Rae1-Nup98 ($K_d$ = 3.92 μM, Fig. 3a, j), confirming that the negative charge of residue 56 is vital to ORF6-Rae1-Nup98 interaction. Collectively, results of these experiments provide mechanistic insights into the role of ORF6 residue 56 in binding Rae1-Nup98 that correlate with ORF6-mediated antagonistic activity.

It was previously reported that the positively charged groove on the rim of Rae1 β-propellers, in which VSV M, MHV68 ORF10 and *Sarbecovirus* ORF6 bind, is a putative RNA binding site of Rae1-Nup98.Therefore, this region is also known as the RNA binding groove[19,22]. Because VSV M can dislocate RNA from Rae1-Nup98 via competitive binding at the RNA binding groove, we speculated that *Sarbecovirus* ORF6 may adopt a similar mechanism. As anticipated, we found that both SARS-CoV-2 and SARS-CoV ORF6 CTTs (C3 peptides) could dislocate single-stranded (ss) RNA from the Rae1-Nup98 complex in a concentration-dependent manner (0–64 μM) in our electrophoretic mobility shift assays (EMSA, Fig. 3k, l). Further, our mutagenesis studies demonstrated that several ORF6 residues important for binding the Rae1-Nup98 complex were also important for ssRNA dislocation from the complex (Fig. 3m). Whereas ORF6 CTT mutants Q56A, E56Q, P57A, E59A, I60A and D61A lost their ability of dislocating ssRNA from Rae1-Nup98 to various extents, mutant M58 was completely unable to compete with ssRNA for binding Rae1-Nup98 (Fig. 3m). Additionally, considerably higher concentrations of VSV M-NTE or MHV68 ORF10-CTT peptides (100–800 μM) were needed to achieve complete RNA dislocation comparing to that for *Sarbecovirus* ORF6 CTT peptides (Supplementary Fig. 4a, b). These results are consistent with our ITC assays showing that SARS-CoV-2 and SARS-CoV ORF6 CTT bound Rae1-Nup98 with higher affinity than that of VSV M-NTE and MHV68 ORF10-CTT peptides.

To validate the structural and biophysical characterizations of ORF6-derived peptides and to understand the function of ORF6 as an intact protein in cells, we studied the interaction between ORF6 and Rae1-Nup98 using co-immunoprecipitation (Co-IP). The Co-IP results revealed that both SARS-CoV-2 and SARS-CoV ORF6 interacted with Rae1-Nup98, while introducing mutations to *Sarbecovirus* ORF6, reduced its binding affinity to Rae1-Nup98 to different degrees (Fig. 4a), which agreed with our ITC results. M58A nearly abolished the binding of SARS-CoV-2 ORF6 to Rae1-Nup98, therefore this key methionine was also essential for their interaction in cells. While I60A and P57A mutations impaired the binding of SARS-CoV-2 ORF6 to Rae1-Nup98, D61A and E59A had only minor effects (Fig. 4a).

Residues 46 and 56 of SARS-CoV-2 ORF6 were identified as the determinants for its immunosuppressive activity[14], although their precise functions remain unknown. While SARS-CoV-2 ORF6 contains E46, residue 46 of SARS-CoV ORF6 is a lysine. Therefore, we swapped residue 46 between these two *Sarbecovirus* ORF6 proteins and analyzed their binding affinities to Rae1-Nup98. While E46K severely undermined the binding of SARS-CoV-2 ORF6 to Rae1-Nup98, K46E enhanced the binding of SARS ORF6 to Rae1-Nup98 (Fig. 4a). The negative charge of residue 46 favored the binding of ORF6 to Rae1-Nup98, which explains the role of E46 in governing ORF6-mediated immunosuppressive activity. As for ORF6 residue 56, Q56A reduced the binding of SARS-CoV-2 ORF6 to Rae1-Nup98, and E56Q of SARS-CoV ORF6 had a similar effect (Fig. 4a). These results are again consistent with the ITC experiments (Fig. 3a). It should be noted that full length MHV68 ORF10 and VSV M exhibited stronger interaction with Rae1-Nup98 compared with that of *Sarbecovirus* ORF6 and its mutants, which was inconsistent with the ITC results shown in Fig. 3. One plausible reason is that full length MHV68 ORF10 and VSV M have additional binding interface with Rae1-Nup98 in addition to the CTT and NTE domains, which were also reported by other studies[19,22]. To verify the ITC results shown in Fig. 3, we constructed ORF6 NTE/ORF10 CTT chimeras (Supplementary Fig. 5b), in which *Sarbecovirus* ORF6

CTT was replaced by MHV68 ORF10 CTT, and we performed Co-IP experiments with those chimeras. As anticipated, SARS-CoV-2 ORF6 NTE / ORF10 CTT (CoV-2 NTE/CTT) and SARS-CoV ORF6 NTE / ORF10 CTT(CoV NTE/CTT) hybrids exhibited weaker interaction with Rae1-Nup98 than that of either *Sarbecovirus* ORF6 or MHV68 ORF10 (Supplementary Fig. 5c). Combining Co-IP and ITC results, we provided compelling evidence that *Sarbecovirus* ORF6 CTT is vital for its interaction with Rae1-Nup98 in cell, which correlated with ORF6-mediated immunosuppressive activity.

To reveal the molecular mechanism of ORF6 in blocking nucleo-cytoplasmic trafficking in cells, we used a green fluorescence protein (GFP) expression plasmid as a reporter for measuring mRNA nuclear export. Overexpressing wild-type SARS-CoV-2 ORF6 significantly downregulated GFP expression level (Fig. 4b, c), indicating that ORF6 restricted GFP mRNAs nuclear transport. The ORF6 mutations that reduced its binding to Rae1-Nup98 (as identified in ITC and Co-IP) also impaired its ability of inhibiting GFP expression to different levels (Fig. 4b, c). Of note, ORF6 mutations M58A, E46K, I60 and E56Q nearly abolished its ability in inhibiting GFP expression. By contrast, ORF6 mutations that marginally affected binding to Rae1-Nup98, such as D61A and E59A, elicited little effects on inhibiting GFP expression.

A comparison of the two *Sarbecovirus* ORF6 proteins revealed that SARS-CoV-2 ORF6 was more potent in inhibiting GFP expression than SARS-CoV ORF6, which correlated with their binding affinities to Rae1-Nup98 and immunosuppressive activities[13]. Notably, both MHV-68 ORF10 and VSV-M exhibited higher activity in inhibiting GFP expression than did *Sarbecovirus* ORF6. This is likely caused by their larger binding interfaces with Rae1-Nup98 than ORF6. Western blotting experiments confirmed that GFP expression was consistent with the fluorescence microscopy results (Supplementary Fig. 5a).Furthermore, consistent with our Co-IP results (Supplementary Fig. 5c), SARS-CoV-2 ORF6 NTE / ORF10 CTT and SARS-CoV ORF6 NTE / ORF10 CTT hybrids that lost interaction with Rae1-Nup98 caused less reduction of GFP expression compared to either *Sarbecovirus* ORF6 or MHV68 ORF10 (Supplementary Fig. 5d–f).

To verify that the observed GFP expression inhibition was caused by ORF6-mediated imprisonment of GFP mRNA in the nucleus, we quantified the nuclear to cytoplasmic (Nu/Cyto) ratio of GFP transcripts as described previously[22]. Overexpressing SARS-CoV-2 and SARS-CoV ORF6 proteins resulted in the accumulation of GFP mRNAs in the nucleus, as indicated by a high Nu/Cyto ratio of GFP transcripts (Fig. 4d). By contrast, the ORF6 mutants that lost the binding affinity to Rae1-Nup98 failed in trapping GFP mRNA in the nucleus. SARS-CoV-2 ORF6 E46K and SARS-CoV ORF6 E56Q (i.e., mutants that lost anti-IFN activity) could not block GFP mRNA nuclear export (Fig. 4b–d). Taken together, these results provide experimental evidence that ORF6-mediated blocking of mRNA nuclear export and inhibition of GFP expression depend on the binding of ORF6 to the Rae1-Nup98 complex. Disrupting this interaction impairs the function of ORF6.

In addition to blocking mRNA nuclear export, SARS-CoV-2 and SARS-CoV ORF6 proteins block STAT1 nuclear import and suppress the IFN-signaling pathway[13,14,17,28,29]. Therefore, we investigated the inter-molecular interactions implicated in the ORF6-mediated inhibition of STAT1 import. Overexpressing SARS-CoV-2 and SARS-CoV ORF6 dramatically inhibited the IFN-sensitive response element-driven luciferase activity triggered by IFN-α and IFN-β, while ORF6 mutants that lost the binding affinity to Rae1-Nup98 also lost their inhibitory activity to different extents (Fig. 4e, f). Specifically, ORF6 harboring the M58A and E46K mutation exhibited the strongest loss of inhibitory activity. The activity of the M58A mutant was similar to that of the negative control (i.e., empty vector). These results indicate that residues M58 and E46 of SARS-CoV-2 ORF6 are essential for its antagonistic activity against IFN-signaling. Finally, we investigated the ability of *Sarbecovirus* ORF6 in blocking STAT1 nuclear import by measuring the distribution of STAT1 in cytoplasmic and nuclear fractions. While IFN-β treatment induced

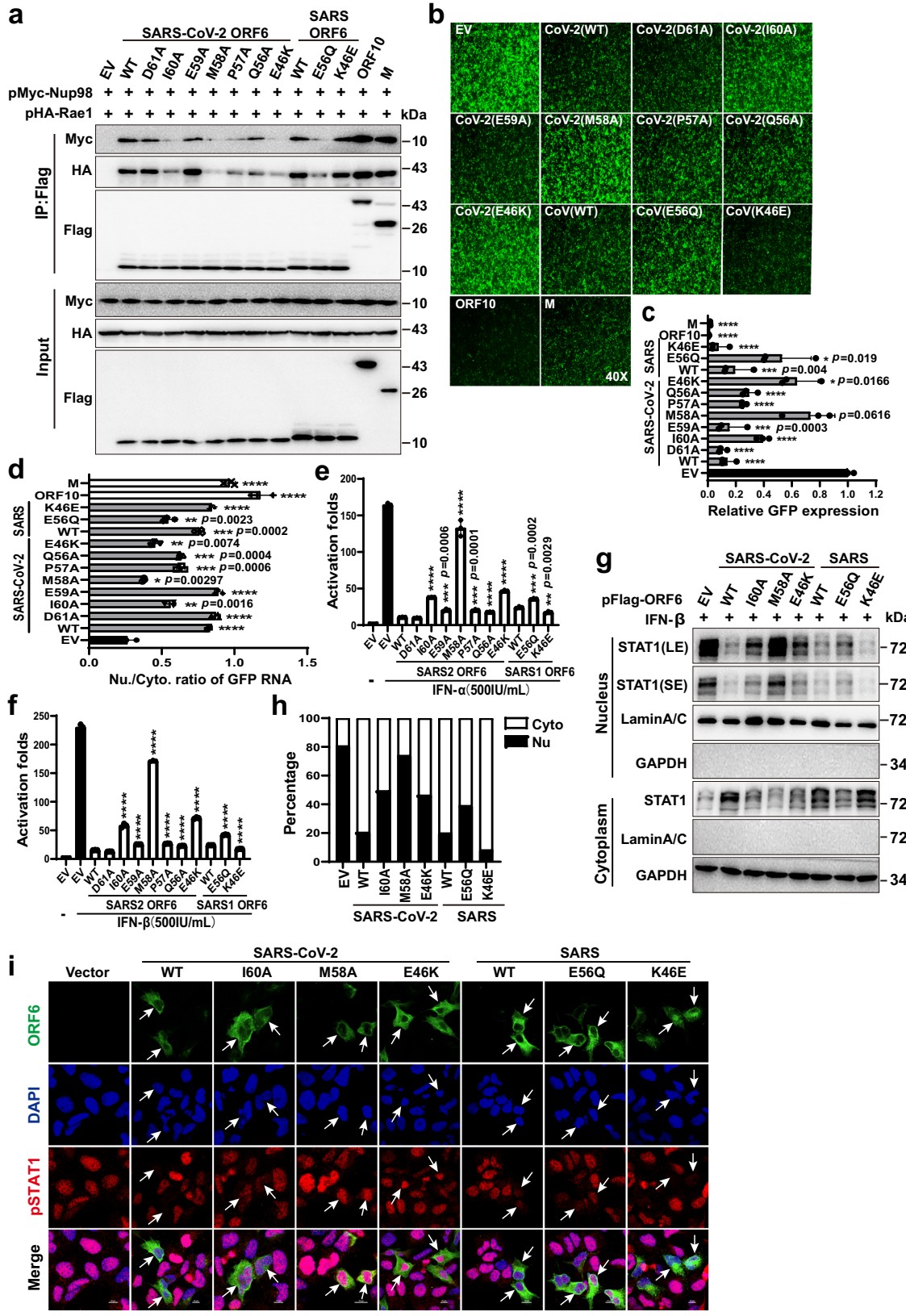

STAT1 nuclear import, overexpressing ORF6 blocked STAT1 translocation into the nucleus. Consistent with results from other experiments in this study, ORF6 mutants that lost the binding affinity to Rae1-Nup98 also lost their inhibitory activity (Fig. 4g, h).In addition, we analyzed phosphor-STAT1 translocation from the cytoplasm to the nucleus by confocal microscopy. Consistent with the results shown in

Fig. 4g, IFN-β treatment triggered endogenous phosphor-STAT1 nuclear translocation in empty vector-transfected cells, while nuclear import was impaired in cells expressing either SARS-CoV-2 or SARS-CoV ORF6. Importantly, the ability of loss-of-interaction mutants of ORF6 to block phosphor-STAT1 translocation to the nuclei was severely impaired (Fig. 4i).

**Fig. 4 | Interactions with Rae1-Nup98 are important for ORF6-mediated disruption of nucleocytoplasmic transport. a** HEK-293T cells were co-transfected with the indicated plasmids, and 48 h later, Co-IP analysis was performed to examine the interaction between Rae1-Nup98 and wild-type (WT) or mutant viral proteins. Results shown are representative of three independent experiments. **b** HEK-293T cells were transfected with pEGFP-C1 and individual plasmid expressing WT or mutant ORF6, ORF10, or M. GFP expression were analyzed by fluorescence microscopy. All fluorescence images at 40 times magnification (40×). Results shown are representative of three independent experiments. **c** Quantification of the fluorescence intensity in **b** with Image J. Data are representative of three independent experiments and shown as the mean ± SD. (*$p < 0.05$, ***$p < 0.001$, ****$p < 0.0001$; two tailed Student's $t$-test, $n = 3$). **d** The cytoplasm and nuclear RNAs were extracted from transfected cells in **b**, and the levels of GFP transcripts were quantified by RT-PCR. The Nu/Cyto ratio is plotted as the mean ± SD. (*$p < 0.05$, **$p < 0.01$, ***$p < 0.001$, ****$p < 0.0001$; two tailed Student's $t$-test, $n = 3$). **e, f** The effects of ORF6 mutants on ISRE-promoter activation. HEK-293T cells were

transfected with individual plasmid expressing WT or mutant ORF6, ISRE-luc and TK-Renilla reporter plasmids, and 24 h later, treated with IFN-α and IFN-β for 16 h. Dual luciferase reporter assay was conducted. Data are representative of three independent experiments and shown as the mean ± SD. (**$p < 0.01$, ***$p < 0.001$, ****$p < 0.0001$; two tailed Student's $t$-test, $n = 3$). **g** The effects of ORF6 mutants on STAT1 nuclear translocation. HEK-293T cells were transfected with individual plasmid expressing WT or mutant ORF6, and 24 h later, the cells were treated with IFN-β (500 IU/ml) for 1 h. The nuclear and cytoplasmic fractions were separated for Western blotting analysis. Results shown are representative of three independent experiments. **h** Quantification of the expression levels of STAT1 in nuclear and cytoplasmic fractions in **g** by Image J. **i** Subcellular localization of pSTAT1. HEK293 cells were transfected with individual plasmid expressing WT, mutant Flag-tagged ORF6, or empty vector (EV). 24 h post-transfection, cells were treated with IFN-β (500 IU/ml) for 1 h. ORF6 and endogenous pSTAT1 were analyzed by confocal microscopy. Scale bars, 10 μm. Results shown are representative of three independent experiments. Source data are provided as a Source data file.

Although previous studies described the role of ORF6 in disrupting nucleocytoplasmic trafficking[28,30], the precise molecular mechanism adopted by ORF6 to disrupt nucleocytoplasmic trafficking remains unclear, e.g., ORF6 mediated STAT1 nuclear import blockade. Following interferons (IFNs) stimulation, STAT1 is tyrosine phosphorylated and dimerizes by intermolecular SH2-phosphotyrosine interactions. This conformation of the phosphorylated STAT1 dimer is recognized by the import receptor KPNA1(importin-α5), and they subsequently bind with KPNB1(importin-β) to form the STAT1-KPNA1-KPNB1 cargo-receptor ternary import complex[31,32].Next, KPNB1 binds to the FG-repeats region of Nup98[33–36], which mediates docking of the STAT1-KPNA1-KPNB1 complex in the cytoplasm to the nuclear pore[32,35,37–39]. Previous studies have demonstrated that SARS-CoV-2 ORF6 disrupts the interaction between Nup98 and KPNA1-KPNB1 through binding with Nup98[16].In the current study, we did not find interaction between SARS-CoV-2/SARS-CoV ORF6 CTTs and the GLEBS motif of Nup98, implying that *Sarbecovirus* ORF6 may interact with other regions of Nup98 rather than the GLEBS motif. The FG-repeats of Nup98 are possibly implicated in binding with ORF6 because the FG-repeats are also the binding sites for KPNB1[34,36,37]. Collectively, SARS-CoV-2 ORF6 might disrupt the formation of Nup98 for the KPNA1-KPNB1 complex via competitive binding with Rae1-Nup98 as previously described[16]. An alternative possibility is that the ORF6-Rae1-Nup98 complex forms a steric hindrance during the binding of KPNB1 to Nup98. As such, the docking of the STAT1-KPNA1-KPNB1 complex to NPC is impaired and ultimately blocks STAT1 nuclear import. Another recent study suggests that ORF6 clogs the nuclear pore via its interactions with Rae1-Nup98, thereby preventing bidirectional nucleocytoplasmic transport[13]. In supporting this model, our structural and biochemical characterizations demonstrate that *Sarbecovirus* ORF6 proteins target on the RNA-binding groove in the Rae1-Nup98 complex and dislocate ssRNA from the complex, which provides evidence for the role of *Sarbecovirus* ORF6 in blocking RNAs nuclear export. Together, we provide here a wealth of experimental evidence demonstrating that binding of ORF6 to Rae1-Nup98 is a fundamental step for blocking nucleocytoplasmic trafficking. Blocking nucleocytoplasmic trafficking ultimately results in innate immunity suppression that facilitates CoV infection. Importantly, this work identifies key determinants in the ORF6 CTT that govern its antagonistic functions.

While our results support that *Sarbecovirus* ORF6 may functions as a "gate-keeper" that forms a steric hindrance by binding to the Rae1-Nup98 complex on the cytoplasmic side of the NPC, a recent study showed that overexpressing ORF6 in cells could displace Rae1-Nup98 from the NPC and also reduce the size of nucleus[40]. However, it remains unclear whether the reduction of nucleus size was caused by the dissociation of Rae1-Nup98 from the NPC. Future structural characterization of the NPC bound to intact ORF6 would better clarify mechanisms underlying ORF6-mediated blockade of nucleocytoplasmic transport.

In summary, this study provides structural basis for the hijacking of the cellular nucleocytoplasmic transport machinery by *Sarbecovirus* ORF6 proteins. Our results reveal atomic details for binding of SARS-CoV-2 and SARS-CoV ORF6 CTT to the Rae1-Nup98 complex, and identify key residues of the CTT that determine the binding affinity of ORF6 to Rae1-Nup98. We prove that the binding affinity of ORF6 to Rae1-Nup98 accounts for its role in nucleocytoplasmic trafficking blockade and IFNs suppression. Intriguingly, a 12-mer short peptide derived from ORF6 CTT sequence exhibited nanomolar binding affinity to Rae1-Nup98, suggesting a starting point for development of novel immunosuppressive drugs.

## Methods

### Reagents and cells

Reagents. All chemicals and reagents used in this study were purchased from Sigma-Aldrich unless otherwise stated. Anti-Flag antibody (Sigma-Aldrich, Cat#: F7425, 1:1000), anti-HA antibody (Sigma-Aldrich, Cat#: H6908, 1:1000), anti-Myc antibody (Sigma-Aldrich, Cat#: SAB4300319, 1:1000), Anti-FLAG M2 affinity Gel (Merck, Cat#: A2220), anti-Lamin A/C (Cell Signaling technology, Cat#: 4777 S, 1:2000), anti-GAPDH (Huaxing bio, Cat#: HX1828, 1:2000), anti-STAT1 (Cell Signaling technology, Cat#: 14994 S, 1:1000), anti-p-STAT1 (Cell Signaling technology, Cat#: 9177 S, 1:100), Goat anti-Mouse IgG (H + L) Highly Cross-Adsorbed Secondary Antibody, Alexa Fluor 488 (Invitrogen, Cat#: A11029, 1:500), Goat anti-Rabbit IgG (H + L) Cross-Adsorbed Secondary Antibody, Alexa Fluor 555 (Invitrogen, Cat#: A21428, 1:500).

Cells. HEK-293T and HEK293 cells were cultured in DMEM supplemented with 1% penicillin–streptomycin and 10% fetal bovine serum. Cell lines used in this study were not found in the BioSample database of commonly misidentified cell lines provided by the International Cell Line Authentication Committee (ICLAC). Cell lines were authenticated by ATCC and were routinely tested for mycoplasma contamination every 3 months.

### Plasmid construction

The genes of human RNA export factor one (*Rae1*; residues 31–368; named by *Rae1*$_{31-368}$), human nucleoporin 98(Nup98) Gle2-binding sequence (GLEBS) motif (residues 157–213; denoted by *Nup98*$_{157-213}$), SARS-CoV-2 ORF6, SARS-CoV ORF6 and VSV M were synthesized and codon optimized for expression in insect cells and 293 T cells (Supplementary Table 4). The *Rae1* and *Nup98*$_{157-213}$ were amplified by PCR and cloned into a pFastBac Dual vector (Invitrogen) for co-expression. While the gene of *Nup98*$_{157-213}$ was inserted to the ORF1 (between BamH I and Hind III restriction sites)with a TEV-cleavable N-terminal His tag and the gene of *Rae1*$_{31-368}$ was inserted to the ORF2 (between XhoI and KpnI restriction sites) without a tag as previously described[19,22,25]. In addition, pHA-Rae1, pMyc-Nup98, pFlag-M and pFlag-ORF6 plasmids were cloned into pCMV-HA, pCMV-Myc and

pcDNA3.1, respectively (Supplementary Table 5). The pFlag-ORF10 have been constructed previously[22].

## Protein expression and purification

All of the recombinant proteins were expressed in insect cells using the Bac-to-Bac Baculovirus Expression System (Invitrogen).One liter Sf21 cells were infected with 30 ml recombinant baculovirus at 22 °C, 2 days after infection. The cells were harvested by centrifugation and the cell pellet was resuspended and lysed by ultrasonication in buffer containing 50 mM Tris-HCl, pH = 8.0, 150 mM NaCl, 10 mM imidazole, 10 mM β-mercaptoethanol and 1 mM PMSF. The resulting mixture was clarified by centrifugation, and the supernatant was passed through Ni-NTA resin pre-equilibrated with the lysis buffer. The target protein was stripped by elution buffer containing 300 mM imidazole. Subsequently, the N terminal His-tag was removed by TEV protease digestion and re-loaded to Ni-NTA resin to remove His tag, the flowthrough containing nontagged target protein was collected and subjected to a HiTrap Q HP column (GE Healthcare) equilibrated with the buffer containing 20 mM Tris-HCl, pH = 8.0, 75 mM NaCl, and eluted with a 10–1000 mM NaCl gradient. Under this condition, Rae1 and Nup98 does not bind to HiTrap Q HP column but further removed non-specifically bound proteins. The flowthrough fraction containing the target protein was finally purified by size-exclusion chromatography using the Superdex 200 10/300 column (GE Healthcare) pre-equilibrated with a storage buffer containing 20 mM Tris pH = 8.0, 100 mM NaCl.

## Crystallization and structure determination

To crystalize Rae1$_{31-368}$-Nup98$_{157-213}$with ORF6 peptide complex, the Rae1$_{31-368}$-Nup98$_{157-213}$ was concentrated to 1 mg/ml before adding SARS-CoV-2 and SARS-CoV ORF6 peptides. The complexes of protein:peptide were reconstituted by incubating protein and peptide at a molar ratio of 1:4. After incubating at 4 °C overnight, the mixture was concentrated to ~5 mg/mL. The concentrated complex was crystallized by mixing 1 μl protein and 1 μl reservoir buffer containing 0.1 M Bis-Tris pH = 6.5, 45% Polypropylene glycol P 400 in a hanging drop vapor diffusion system at 18 °C. The crystals appeared in 3 days and grew to maximum size in about 1 week. For cryoprotection, the crystals were briefly soaked in the reservoir buffer supplemented with 20% ethylene glycol before flash-frozen in liquid nitrogen. Complete X-ray diffraction data were collected at 100 K at the Shanghai synchrotron radiation facility (SSRF) beamline BL10U2 and BL19U1, Shanghai China. X-ray intensities were processed using the XDS package[41]; the structure was solved by molecular replacement using the software Phaser MR in CCP4 (PDB 4OWR was used as the searching model) to yield interpretable initial electron density map[42]. Manual model building was performed using the software Coot and PHENIX was used to finally refine the structure[43]. The statistics of data collection and structure refinement are summarized in Supplementary Table 2. Structural presentations were prepared using the software Pymol.

## Size-exclusion chromatography

The purified Rae1$_{31-368}$-Nup98$_{157-213}$ heterodimer proteins were loaded to Superdex 200 10/300 GL column (GE healthcare) pre-calibrated using molecular weight standards: γ-globulin (158 kDa), ovalbumin (45 kDa), myoglobin (17 kDa) and vitamin B12 (1.35 kDa) in buffer containing 20 mM Tris pH = 8.0, 100 mM NaCl. Size-exclusion chromatography profile of Rae1·Nup98 complex was analyzed by Graphpad Prism. The elution fractions were further analyzed with SDS-PAGE.

## Isothermal titration calorimetry

An isothermal titration calorimetry (ITC) assay was conducted using a MicroCal iTC200 calorimeter (MicroCal, USA) at 25 °C as previously described[10,44,45]. Both the Rae1$_{31-368}$-Nup98$_{157-213}$ heterodimer proteins and ORF6 peptides were dissolved in the same buffer (20 mM Tris-HCl,

pH = 8.0, 100 mM NaCl). The concentrations of peptides in the syringe were between 0.5 and 1 mM and the concentration of proteins in the sample cell were 0.03 mM. The ORF6 peptides were titrated into Rae131$_{-368}$-Nup98$_{157-213}$ heterodimer proteins with a 120-s interval between injections using a stir rate of 400 rpm. We used 18 consecutive 2 μl injections of peptides to determine affinities and thermodynamic parameters. A single-site binding model was used for nonlinear curve fitting using Microcal Origin software provided by the manufacturer. ITC experiments were repeated twice for each sample.

## EMSA

The EMSA assay was performed as previously described and slightly modified[19,22]. Briefly, a 10-mer poly (U) ssRNA was synthesized with fluorescein amidite (FAM) label at the 5' end (GenScript). The 0.2 μM ssRNA was first incubated with 2 μM Rae1·Nup98 complex in a buffer containing 10 mM Tris (pH 8.0), 150 mM NaCl, and 0.5 mM TCEP, for 15 min at room temperature. The various peptides were added to the mixture at different concentrations (0–64 μM for ORF6 CTT and mutants, 0–800 μm for ORF10 CTT and M NTE), and incubated for an additional 10 min at room temperature. The reaction samples were loaded onto a 6% native-PAGE gel and run in 45 mM Tris (pH 8.5, titrated with boric acid) buffer at 4 °C. After electrophoresis, the ssRNA was visualized using the fluorescence signal from the FAM label using a Typhoon Trio Variable Mode Imager (GE Healthcare).

## Co-IP and western blotting

HEK-293T cells in 6 cm dish transfected with indicated plasmids were lysed in 600 μl lysis buffer [50 mM Tris-HCl, pH 8.0; 150 mM NaCl, 1% Trition X-100, 1 mM phenylmethanesulfonyl fluoride, EDTA-free protease inhibitor cocktail (Roche 04693)] for 30 min at 4 °C. The supernatant of cell lysates was obtained by centrifugation at 15,870×g for 15 min at 4 °C. 50 μl cell lysate was taken as input, and the rest was incubated with anti-FLAG-M2-conjugated agarose for 5 h at 4 °C. The protein-bound were washed five times with lysis buffer at 4 °C and then boiled in SDS-PAGE loading buffer for western blotting. The samples were analyzed by SDS-PAGE and the indicated antibodies. All western blot images were scanned and collected using MINICHEMI (SAGECREATION).

## Nuclear and cytoplasmic RNA/protein fractionation

Nuclear and cytoplasmic RNA/protein fractionation was extracted according to a method described previously[22]. Briefly, 293 T cells in 12-well plates were collected by trypsinization, and further washed with cold DEPC-treated PBS three times. Then, cells were pelleted at 250 × g for 3 min at 4 °C and resuspended in RSB buffer (10 mM Tris, pH 7.4, 10 mM NaCl, 3 mM MgCl$_2$, 1 mM DTT), incubated on ice for 5 min, followed by centrifugation at 250 × g for 3 min. The cell pellet was resuspended by pipetting with 100 μL lysis buffer RSBG40 (10 mM Tris, pH 7.4, 10 mM NaCl, 3 mM MgCl$_2$, 10% glycerol, 0.5% NP-40, 1 mM DTT). The nuclei were pelleted at 250×g for 5 min. The supernatant was saved as the cytoplasmic fraction, and centrifuged twice at 15,870×g for 10 min at 4 °C to remove any nuclei or cellular debris. The nuclei were resuspended with 1 ml RSBG40D buffer by gently pipetting, incubated on ice for 5 min, and then pelleted at 1500 rpm for 3 min. After washing with RSBG40D buffer three times, the nuclei were collected as the nuclear fraction. After nuclear and cytoplasmic fractions were separated, the extracted RNA or proteins was carried out for RT-PCR and western blotting respectively.

## Reverse transcription and real-time PCR

The mRNA was reverse-transcribed to cDNA according to Genomic DNA Eraser Reverse Transcription Kits (Takara, catalog no. RR047A). Briefly, 1 μg of RNA was treated with DNA eraser enzyme to remove genomic DNA, and then cDNA was synthesized using PrimeScript RT enzyme mix and RT primer mix. 100 ng RNA were subjected to Real-

time PCR (Quantstudio 7, Applied Biosystems) to determine the GFP RNA transcript copy number, and the primers for RT-PCR were designed according to Gong et al.[20].

## Luciferase reporter assay

HEK-293T cells were transfected with ISRE-firefly luciferase reporter plasmid, pRL-TK (renilla luciferase) and the indicated expression plasmids. After 24 h transfection, cells were treated with IFN-β and IFN-α (500 IU/mL) for 16 h, and luciferase activity was measured. The dual luciferase assay kit (Promega E1960) was used to perform luciferase assays, and GloMax-Multi JR detection system (Promega) was used to quantify luciferase activity.

## Immunofluorescence assay (IFA)

HEK293 cells were seeded in 24 well plates and transfected with Flag-tagged SARS-CoV-2 and SARS-CoV ORF6 and mutants or empty vector (EV). After a 24 h transfection, cells were stimulated with IFN-β (500 IU/ml) for 1 h. Subsequently, cells were fixed with 4% paraformaldehyde and permeabilized with 0.2% Triton X-100 in phosphate-buffered saline (PBS). Cells were blocked and incubated with anti-FLAG and anti-pSTAT1 primary antibodies overnight at 4 °C. The fluorophore-conjugated secondary antibodies were diluted 1:500, and nuclei were visualized with DPAI staining. The antibodies used in this assay were as follows: FLAG antibody from Sigma-Aldrich (1:200, Cat# F7425); and p-STAT1 from Cell Signaling technology (1:100, Cat# 9177 S). Fluorescence images were obtained using a confocal microscope (A1R +, Nikon) and analyzed with NIS-Elements viewer 4.20.

## Reporting summary

Further information on research design is available in the Nature Research Reporting Summary linked to this article.

## Data availability

The atomic coordinates and structure factors have been deposited in the Protein Data Bank under the accession codes: 7F60 (SARS-CoV-2 ORF6 in complexed Rae1-Nup98) and 7F90 (SARS-CoV ORF6 in complexed Rae1-Nup98). Publicly available protein atomic models with the following PDB code was used in the study: 4OWR. Other data are available from the corresponding author upon reasonable request. Source data are provided with this paper.

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

## Acknowledgements

We thank the staffs of BL19U1 beamline of National Facility for Protein Science in Shanghai (NFPS), the staff of the BL17U1 beamline and the staff of the BL10U2 beamline at the Shanghai Synchrotron Radiation Facility for assistance in data collection. We thank Prof. Xiaoyun Ji and Dr. Tinghan Li from Nanjing University for the help with EMSA assay. This work was supported by the Chinese Academy of Medical Sciences (CAMS) Innovation Fund for Medical Sciences (2021-I2M-1-037 – S. C and X. G.) and the National Key Research and Development Program of China (2016YFD0500300 – S. C. and 2019YFC0840602 – X. G.); National Natural Science Foundation of China (No. 81971985 – X. G., 81772207 – S. C., 81572005 – S.C, 81921005 – H. D., 31900131 – H. T.); The Chinese Academy of Sciences (the Strategic Priority Research Program XDB37030205 – H. D., and KJZD-SW-L05 – H. D.); The Ministry of Science and Technology (2021YFA1300802 – H. D.); Beijing Natural Science Foundation (No. 5222024 – H. T).

## Author contributions

S. C., H. D. and X. G. designed the study. S. C., H. D., H. T. and X. G. solved the structure and wrote the paper. X. G., H. T., K. Z., Q. L., L. W., B. Q. and W. H. performed experiments, analyzed the data and revised the paper. All authors reviewed the results and approved the final version of the manuscript.

## Competing interests

Patents protecting the design and application of the ORF6-derived peptides (including its derivatives) described in this paper is pending by the authors of this paper, Institute of Pathogen Biology, Chinese Academy of Medical Sciences and Peking Union Medical College (Beijing, 100730, P. R. China) and CAS Key Laboratory of Infection and Immunity, CAS Center for Excellence in Biomacromolecules, Institute of Biophysics, Chinese Academy of Sciences, Beijing 100101, China; University of Chinese Academy of Sciences, Beijing 100049, China.
