## [Peer Review File · Nature Communications]

Structural Basis for Sarbecovirus ORF6 Mediated Blockage of Nucleocytoplasmic TransportReviewers' Comments:

Reviewer #1:

Remarks to the Author:

The manuscript by Gao et al reports the structural basis for the recognition of Rae1-Nup98 by SARS-CoV and SARS-CoV-2 Orf6 protein. Orf6 is a key interferon antagonist, and it has been implicated in blocking nuclear import of STAT as well as nuclear export of mRNA. The crystal structures of Orf6-Rae1-Nup98 reported here reveal the molecular details for the interaction between Orf6 and Rae1-Nup98. The authors examined the contribution of individual Orf6 residues to Rae1-Nup98 binding affinity and the disruption of mRNA nuclear export as well as STAT1 nuclear import. The results showed that Met58 is critical for Orf6's inhibitory functions, which is confirmatory of published studies. Several Orf6 residues flanking Met58 also contribute to Rae1-Nup98 recognition. Overall, this work provides an important piece of information on how Orf6 targets Rae1-Nup98, but does not significantly extend the current working model of Orf6-mediated activities.

Here are some specific points.

1. The statement "P57 governs the orientation of binding" in the abstract (detailed description in page 6) is incorrect. The N' and C' of VSV M protein are incorrectly labeled in Fig 2E. In fact, Orf6, MHV-68 Orf10, and VSV M bind to Rae1 in the same N' to C' orientation.

2. Regarding the Orf6 C-terminal YP extension (line 156-165), the authors reasoned that removing YP from SARS-CoV Orf6 introduced extra negative charge from the terminal carboxyl group, thereby increasing the binding affinity to Rae1. If this were true, adding YP extension to SARS-CoV-2 Orf6 would decrease the binding affinity, which is different from the observation here.

3. Can the authors compare their structures with the recently published Orf6-Rae1-Nup98 structures by Li et al in Front Mol Biosci?

4. The manuscript needs careful proofreading. Below are a few examples.

Figure 3B to 3T, dissociation constants are all labeled as kD in the figure and the legend.

Line 28: "the C-terminal (CTT) of SARS-CoV-2". Orf6 is missing.

Line 349: "The complexes of protein:peptide were reconstituted by incubating protein and peptide were mixed at..."

Line 557: "E-H. ITC titrations of between...."

Reviewer #2:

Remarks to the Author:

This manuscript offers structures and binding affinities of the CTT of SARS-CoV and SARS-CoV-2 orf6 with the Rae1-Nup98 complex, an important target of this accessory gene in the sarbecoviruses.

The authors recover additional orf6 mutations associated with increased and decreased nucleocytoplasmic trafficking. Many of the other assays are standard and confirmatory for prior work that has been done on the orf6-rae1-nup98 complex. The main addition is the structure which has only been imputed to date as well as the direct binding assays and estimations of Kd, which are both commendable and of significance to the field.

Minor comments.

-axis in Figure 3A should be labeled.

-the reduced affinity of MHV68 and VSV M compared to SARS/SARS2 takes away from the overall

argument about binding affinity influencing strength of repression. one possible route to explore this further would be to just express the CTT or NTE of MHV68/VSV M (or make a chimeric orf6 N-terminal/MHV68 CTT hybrid) and see if these proteins cause less of a reduction of GFP expression compared to orf6.

-would also like to see MHV68/VSV M on the blot in fig 4A.

-would also like to see the authors add more interpretation of results in the discussion regarding the following..1) speak to how orf6 is capable of blocking stat1 nuclear import from their structure 2) make it more quantitative (e.g., confocal images) to be better able to see if binding affinity also correlates with strength of nuclear import blockade 3) discuss whether the orientation of the viral protein-RAE1 (fig 2E) influences ability to block nuclear import.

rather than fig 4B-D, a more direct measure of binding affinity vs RNA export would be more interesting. specifically, i'm thinking if they could look at degree of RNA dislocation from the rae1/nup98 complex across their different constructs (something like fig 3 in <https://www.ncbi.nlm.nih.gov/pmc/articles/PMC4078809/>), that could better make their argument of binding affinities stronger and more directly connected to mechanism.

Point-to-point responses to reviewers' comments

Reviewer #1 (Remarks to the Author):

The manuscript by Gao et al reports the structural basis for the recognition of Rae1-Nup98 by SARS-CoV and SARS-CoV-2 Orf6 protein. Orf6 is a key interferon antagonist, and it has been implicated in blocking nuclear import of STAT as well as nuclear export of mRNA. The crystal structures of Orf6-Rae1-Nup98 reported here reveal the molecular details for the interaction between Orf6 and Rae1-Nup98. The authors examined the contribution of individual Orf6 residues to Rae1-Nup98 binding affinity and the disruption of mRNA nuclear export as well as STAT1 nuclear import. The results showed that Met58 is critical for Orf6's inhibitory functions, which is confirmatory of published studies. Several Orf6 residues flanking Met58 also contribute to Rae1-Nup98 recognition. Overall, this work provides an important piece of information on how Orf6 targets Rae1-Nup98, but does not significantly extend the current working model of Orf6-mediated activities.

Response:

We thank this reviewer for the comments. In this study, we provide a structural basis for the hijacking of the cellular nucleocytoplasmic transport machinery by two important Sarbecovirus ORF6 proteins, and identify key residues of ORF6 CTT that determine binding affinity of ORF6 to Rae1-Nup98. Furthermore, we demonstrate that the binding affinity of ORF6 to Rae1-Nup98 is relevant to its role in nucleocytoplasmic trafficking blockade and IFNs suppression. Additionally, in the revised manuscript, we added new results showing a correlation of ORF6/Rae1-Nup98 binding affinity with RNA disassociation from the Rae1-Nup98 complex. By combining these findings, we propose a strategy for using ORF6 CTT peptide as a potential immunosuppression agent.

Here are some specific points.

1. The statement "P57 governs the orientation of binding" in the abstract (detailed description in page 6) is incorrect. The N' and C' of VSV M protein are incorrectly labeled in Fig 2E. In fact, Orf6, MHV-68 Orf10, and VSV M bind to Rae1 in the same N' to C' orientation.

Response:

We apologize for the incorrect description and the labelling in Fig 2E. We corrected these sentences in main text (page 6, line 88-91) and the labeling in the revised Fig 2E (see revised Fig 2e).

2. Regarding the Orf6 C-terminal YP extension (line 156-165), the authors reasoned that removing YP from SARS-CoV Orf6 introduced extra negative charge from the terminal carboxyl group, thereby increasing the binding affinity to Rae1. If this were true, adding YP extension to SARS-CoV-2 Orf6 would decrease the binding affinity, which is different from the observation here.

Response:

We thank this reviewer for pointing out this problem. Our ITC results did indicate that while adding a YP extension did not affect binding affinity of SARS-CoV-2 ORF6 to Rae1-Nup98, it decreased the binding affinity of SARS-CoV ORF6. We therefore rewrite the reasoning in main text (page 8, line 131-134). We believe that the different role of YP in the binding of SARS-CoV-2 ORF6 C3 and SARS-CoV ORF6 C3 peptides to the Rae1-Nup98 complex implies that YP might modulate binding in synergy with other residues specific to SARS-CoV ORF6, but not to SARS-CoV-2 ORF6.

3. Can the authors compare their structures with the recently published Orf6-Rae1-Nup98 structures by Li et al in Front Mol Biosci?

Response:

We thank the reviewer's valuable suggestion. We made comparison between recently published ORF6 CTT-Rae1-Nup98 structures by Li et al in Front Mol Biosci with our structures. We added Supplementary Figure 2 and description of structural comparison in the revised manuscript (page 7, line 101-108).

4. The manuscript needs careful proofreading. Below are a few examples. Figure 3B to 3T, dissociation constants are all labeled as kD in the figure and the legend.

Line 28: "the C-terminal (CTT) of SARS-CoV-2". Orf6 is missing.

Line 349: The complexes of protein:peptide were reconstituted by incubating protein and peptide were mixed at..."

Line 557: "E-H. ITC titrations of between...."

Response:

Our revised manuscript has been proofread and corrected by a native English speaker. All mistakes are modified in accordance with reviewer's suggestion.

The editing certificate can be viewed at the following web address:

[\[https://www.bioedit.com/digital-certificate/view/da0a9a55d935b84ef656abc72c4b3c7156905e81\]](https://www.bioedit.com/digital-certificate/view/da0a9a55d935b84ef656abc72c4b3c7156905e81)

Reviewer #2 (Remarks to the Author):

This manuscript offers structures and binding affinities of the CTT of SARS-CoV and SARS-CoV-2 orf6 with the Rae1-Nup98 complex, an important target of this accessory gene in the sarbecoviruses.

The authors recover additional orf6 mutations associated with increased and decreased nucleocytoplasmic trafficking. Many of the other assays are standard and confirmatory for prior work that has been done on the orf6-rae1-nup98 complex. The main addition is the structure which has only been imputed to date as well as the direct binding assays and estimations of K_d, which are both commendable and of significance to the field.

Response:

We thank this reviewer for summarizing the key contributions of our work and positive comments.

Minor comments.

-axis in Figure 3A should be labeled.

Response:

We added x axis label (folds) in the revised Fig. 3a, and provided details of calculating this value in the legends (page 37, line 2-3).

-the reduced affinity of MHV68 and VSV M compared to SARS/SARS2 takes away from the overall argument about binding affinity influencing strength of repression. one possible route to explore this further would be to just express the CTT or NTE of MHV68/VSV M (or make a chimeric orf6 N-terminal/MHV68 CTT hybrid) and see if these proteins cause less of a reduction of GFP expression compared to orf6.

Response:

We carried out new experiments and added more results to the revised manuscript. The full length MHV68 ORF10 and VSV M actually exhibited a stronger interaction with Rae1-Nup98 than that of full length Sarbecovirus ORF6 (revised Fig 4a). The stronger binding interaction between MHV68 ORF10 and VSV M, and Rae1-Nup98 is because full length ORF10 and M have additional binding interfaces besides the CTT and NTE domain, which were reported by previous studies^{1,2}. For example, the VSV M protein binds to Rae1-Nup98 via both its globular region and an extended finger region (NTE)¹. By following the reviewer's suggestion, we constructed chimeric ORF6 NTE/ORF10 CTT hybrids (see new Supplementary Figure 5b) and performed Co-IP experiments as well as GFP reporter assays. As expected, CoV2 (NTE)/CTT and CoV (NTE)/CTT hybrids exhibited less interaction with Rae1-Nup98 compared to WT ORF6 and ORF10 (see new Supplementary Figure 5c).

Consistent with the Co-IP results, CoV2 (NTE)/CTT and CoV (NTE)/CTT hybrids that reduce interaction with Rae1-Nup98 caused less of reduction of GFP expression than WT ORF6 and ORF10 did (see new Supplementary Figure 5d-f). We added these new results to the revised manuscript (page 12-13, line 225-236 and page 14, line 257-261).

-would also like to see MHV68/VSV M on the blot in fig 4A.

Response:

We thank the reviewer for the suggestion. We included MHV-68 ORF10 and VSV M on the western blotting in revised Fig. 4a.

-would also like to see the authors add more interpretation of results in the discussion regarding the following..1) speak to how orf6 is capable of blocking stat1 nuclear import from their structure 2) make it more quantitative (e.g., confocal images) to be better able to see if binding affinity also correlates with strength of nuclear import blockade 3) discuss whether the orientation of the viral protein-RAE1 (fig 2E) influences ability to block nuclear import.

Response:

We appreciate the Reviewer's suggestions. As requested by the reviewer, the discussion was expanded in the revised manuscript. The new discussion points are noted below:

- 1. Currently, the precise molecular mechanism adopted by ORF6 mediated STAT1 nuclear import blockade remains unclear. Following interferons(IFNs) stimulation, STAT1 is tyrosine phosphorylated and dimerizes by intermolecular SH2-phosphotyrosine interactions, this conformation of the phosphorylated STAT1 dimer is recognized by the import receptor KPNA1(importin- α 5),and subsequently bind to KPNB1(importin- β) to form STAT1-KPNA1-KPNB1 cargo-receptor ternary import complex^{3 4}. KPNB1 binds to the FG-repeats region of Nup98⁵⁻⁸, which mediates docking of STAT1-KPNA1-KPNB1 complex in the cytoplasm to the nuclear pore^{4,7,9-11}. Miorin L et al. have reported that SARS-CoV-2 ORF6 disrupts the interaction of Nup98 with KPNA1-KPNB1 by binding to Nup98¹². However, we found no interactions between the CTTs of SARS-CoV-2/SARS-CoV ORF6 and the GLEBS motif of Nup98, which implied that ORF6 may interact with Nup98 via regions other than GLEBS, possibly FG-repeats region, which is also the binding site for KPNB1. Therefore, as Miorin L et al. suggested that binding of SARS-CoV-2 ORF6 to Rae1-Nup98 could compete with Nup98 for KPNA1-KPNB1. An alternative possibility is that the ORF6-Rae1-Nup98 complex forms steric*

hindrance during KPNB1 binding to Nup98, thus hinder docking of the STAT1-KPNA1-KPNB1 complexes to nuclear pore complex (NPC), which blocks STAT1 nuclear import. Addetia A et al. have proposed that ORF6 clogs the nuclear pore via its interactions with Rae1-Nup98 to prevent bidirectional nucleocytoplasmic transport (nuclear import of protein and nuclear export of mRNA)¹³. In agreement with this model, our structural and biochemical characterizations demonstrate that Sarbecovirus ORF6 proteins target on the RNA-binding groove in Rae1-Nup98 complex and dislocates ssRNA from the complex, which provides some evidences for the role of Sarbecovirus ORF6 in blocking RNAs nuclear export.

Collectively, we propose that Sarbecovirus ORF6 may functions as a “gate-keeper” that forms steric hindrance by binding to the Rae1-Nup98 complex on the cytoplasmic side of the NPC to block STAT1 nuclear import. Therefore, future structural characterization of the NPC bound to intact ORF6 would better clarify mechanisms underlying ORF6-mediated blockade of nucleocytoplasmic transport. We added more discussion in the revised manuscript (page16-17, line 299-324).

2. By following the reviewer’s suggestion, the expression levels of STAT1 in the nucleus and cytoplasm were analyzed and quantified by Image J. The results show that WT ORF6 of SARS-CoV-2 and SARS-CoV efficiently inhibited the nuclear import of STAT1, while the mutants with reduced binding affinity to Rae1-Nup98 showed varying degrees of blockade in the nuclear import of STAT1 (see revised Fig. 4h). We also performed IFA experiments to determine the effect of ORF6 and mutants on nuclear import of endogenous pSTAT1 after IFN- β treatment. IFA results confirmed that binding affinity of ORF6 WT and mutants correlated with strength of p-STAT1 nuclear import blockade (see revised Fig 4i). We have added these new results to the revised manuscript (page15-16, line 290-297).
3. Structural superimposition demonstrated that the N terminal finger of VSV M, the CTT of MHV-68 ORF10 and ORF6 CTT bind within the same groove of Rae1-Nup98 and share a similar conformation. ORF6 CTT/ORF10 CTT bind to the same side of the Rae1–Nup98 complex, while VSV M binds to the opposite side of the Rae1–Nup98 complex. In the Co-IP and nucleocytoplasmic transport assay, we concluded that viral protein-mediated blocking of mRNA nuclear export and inhibition of GFP expression depends on the binding ability of the viral protein to the Rae1-Nup98 complex. The full length MHV-68 ORF10 and VSV M exhibited stronger interaction with Rae1-Nup98 and inhibited GFP expression to a higher extent than that of ORF6. This phenomenon can be explained by the

fact that full length ORF10 and M have other interaction interfaces besides CTT and N terminal finger domain, which have been reported to contribute to Rae1–Nup98 interaction^{1,2}. In conclusion, we proposed that the binding interface but not the orientation of the viral protein-Rae1 influences the ability to block nuclear import.

rather than fig 4B-D, a more direct measure of binding affinity vs RNA export would be more interesting. specifically, i'm thinking if they could look at degree of RNA dislocation from the rae1/nup98 complex across their different constructs (something like fig 3 in <https://www.ncbi.nlm.nih.gov/pmc/articles/PMC4078809/>), that could better make their argument of binding affinities stronger and more directly connected to mechanism.

Response:

We agree with this reviewer that measuring ORF6-mediated inhibition of RNA binding with Rae1-Nup98 would be very interesting and informative. We therefore used EMSA to investigate the degree of RNA dislocation from the Rae1-Nup98 complex across different constructs as the Reviewer suggested. As expected, both SARS-CoV-2 and SARS-CoV ORF6 CTT inhibited RNA binding to the Rae1–Nup98 complex in a concentration-dependent manner (Revised Fig.3k,l), and mutations in ORF6 CTT led to reduced inhibition of RNA binding with the Rae1–Nup98 complex (Revised Fig.3m). A complete loss of RNA binding inhibition was observed for the SARS-CoV-2 ORF6 CTT M58A mutant (Revised Fig.3m). We also noticed that to achieve RNA inhibition, a considerably higher concentration of ORF10 CTT and M NTE peptides than ORF6 CTT (supplementary Fig 4a, b) was required, which was also consistent with our ITC results. These results show that SARS-CoV-2 and SARS-CoV ORF6 CTTs bound Rae1-Nup98 with higher affinity than ORF10 CTT and M NTE peptides. We added a description of these new results in our revised manuscript (page 11, line 185-204).

References

- 1 Quan, B., Seo, H. S., Blobel, G. & Ren, Y. Vesiculoviral matrix (M) protein occupies nucleic acid binding site at nucleoporin pair (Rae1 * Nup98). *Proc Natl Acad Sci U S A* **111**, 9127–9132, doi:10.1073/pnas.1409076111 (2014).
- 2 Feng, H. *et al.* Molecular mechanism underlying selective inhibition of mRNA nuclear export by herpesvirus protein ORF10. *Proc Natl Acad Sci U S A* **117**, 26719–26727, doi:10.1073/pnas.2007774117 (2020).
- 3 Reich, N. C. STATs get their move on. *JAKSTAT* **2**, e27080, doi:10.4161/jkst.27080 (2013).

- 4 McBride, K. M., Banninger, G., McDonald, C. & Reich, N. C. Regulated nuclear import of the STAT1 transcription factor by direct binding of importin- α . *EMBO J* **21**, 1754–1763, doi:10.1093/emboj/21.7.1754 (2002).
- 5 Radu, A., Moore, M. S. & Blobel, G. The peptide repeat domain of nucleoporin Nup98 functions as a docking site in transport across the nuclear pore complex. *Cell* **81**, 215–222, doi:10.1016/0092-8674(95)90331-3 (1995).
- 6 Bayliss, R., Littlewood, T. & Stewart, M. Structural basis for the interaction between FxFG nucleoporin repeats and importin- β in nuclear trafficking. *Cell* **102**, 99–108, doi:10.1016/s0092-8674(00)00014-3 (2000).
- 7 Shen, Q., Wang, Y. E. & Palazzo, A. F. Crosstalk between nucleocytoplasmic trafficking and the innate immune response to viral infection. *J Biol Chem* **297**, 100856, doi:10.1016/j.jbc.2021.100856 (2021).
- 8 Bonifaci, N., Moroianu, J., Radu, A. & Blobel, G. Karyopherin β 2 mediates nuclear import of a mRNA binding protein. *Proc Natl Acad Sci U S A* **94**, 5055–5060, doi:10.1073/pnas.94.10.5055 (1997).
- 9 Fontoura, B. M., Blobel, G. & Yaseen, N. R. The nucleoporin Nup98 is a site for GDP/GTP exchange on ran and termination of karyopherin β 2-mediated nuclear import. *J Biol Chem* **275**, 31289–31296, doi:10.1074/jbc.M004651200 (2000).
- 10 Chook, Y. M. & Blobel, G. Karyopherins and nuclear import. *Curr Opin Struct Biol* **11**, 703–715, doi:10.1016/s0959-440x(01)00264-0 (2001).
- 11 Lott, K. & Cingolani, G. The importin β binding domain as a master regulator of nucleocytoplasmic transport. *Biochim Biophys Acta* **1813**, 1578–1592, doi:10.1016/j.bbamcr.2010.10.012 (2011).
- 12 Miorin, L. *et al.* SARS-CoV-2 Orf6 hijacks Nup98 to block STAT nuclear import and antagonize interferon signaling. *Proc Natl Acad Sci U S A* **117**, 28344–28354, doi:10.1073/pnas.2016650117 (2020).
- 13 Addetia, A. *et al.* SARS-CoV-2 ORF6 Disrupts Bidirectional Nucleocytoplasmic Transport through Interactions with Rael and Nup98. *mBio* **12**, doi:10.1128/mBio.00065-21 (2021).

Reviewers' Comments:

Reviewer #1:

Remarks to the Author:

The revised manuscript has been significantly improved. The authors have addressed the comments raised in the initial review.

Reviewer #2:

None